# Comparison of Different Universal Adhesive Systems on Dentin Bond Strength

**DOI:** 10.3390/ma16041530

**Published:** 2023-02-12

**Authors:** Sandra Brkanović, Eva Klarić Sever, Josipa Vukelja, Anja Ivica, Ivana Miletić, Silvana Jukić Krmek

**Affiliations:** School of Dental Medicine, University of Zagreb, Gunduliceva 5, 10000 Zagreb, Croatia

**Keywords:** dental adhesives, shear strength, dentin

## Abstract

Over the past few decades, adhesive dentistry has advanced significantly. In light of minimal-invasive dentistry, this novel technique advocates a more conservative cavity design that relies on the efficiency of present enamel-dentine adhesives. The study aimed to address the scientific deficit in understanding the long-term bonding performance of universal adhesives and to provide a new clinical solution with desirable bond strength to dentin. The dentin bond strength of three bonding agents, G2-Bond Universal (GC), Clearfil SE Bond (Kuraray), and Scotchbond Universal Plus (3M ESPE), was evaluated following various storage and etching modes. The UltraTester (Ultradent) bond strength testing machine was used to assess shear bond strength. The results showed that thermal cycling and the choice of adhesive system significantly affected the shear bond strength (*p* = 0.018 and *p* = 0.001, respectively). Among the three adhesives, Scotchbond Universal Plus had the lowest bond strength value (mean value = 24.78 MPa), while G2-Bond Universal was found to have desirable shear bond strength to dentin compared to the other adhesives, even after one year in the oral environment (mean value = 35.15 MPa). These findings imply that the HEMA-free universal adhesive G2-Bond Universal is the most effective universal adhesive for clinical practices, particularly when applied in the self-etch mode.

## 1. Introduction

Today’s restorative dentistry tends toward total removal of carious tissue, while avoiding removal of healthy tooth structure for increased mechanical retention. GV Black’s principal “extension for prevention” is no more justified and has been replaced with a modern “minimally-invasive” approach [1]. Adhesion is the cornerstone of modern restorative dentistry. Dental bonding systems serve as an intermediary substance that binds restorative materials to a hard dental tissue and improves retention, marginal sealing, and tooth-restoration interface resistance [2]. Polymerization stress, however, is the biggest obstacle in the way of composite materials and a major factor in the clinical failure of present-day adhesion methods [3]. Polymerization shrinkage happens during the curing process of composite resins when monomers combine together and make polymers, resulting in a lower total volume. Inner contraction tensions and stresses at the edges of restoration may result from this [4]. If the bond strength is compromised by contraction forces, this results in marginal failure, or postoperative sensitivity, marginal microleakage and marginal staining. The success of restorative procedure depends on the efficacy of adhesive materials to bond. Accordingly, dental adhesives have become one of the essential materials in restorative dentistry and one of the engaging bio-materials in Health Sciences [3]. 

Research efforts of simplifying multistep dental adhesives and making them user friendly have led to the development of “universal” adhesives. Because of their lower toxicity and their ability to be used in both self-etching and etch-and-rinse procedures as well as their flexibility and the number of application steps universal adhesives have become popular in dentistry [5]. After Kuraray Noritake Dental’s patent for 10-MDP (10 methacryloyloxydecyl dihydrogen phosphate) expired in 2003, manufacturers began to explore the use of 10-MDP and other phosphoric acid esters for innovative adhesive compositions. In 2012, the first universal adhesive was released for sale in Japan. It was called Scotchbond Universal Adhesive and was manufactured by 3M Oral Care in St. Paul, Minnesota. Later, universal adhesives were launched that can be used with resin luting cements [6], a variety of substrates without surface treatment [7], shortened treatment times [8] or a variety of surface moistures of enamel and dentin surfaces [9]. Despite this intriguing versatility, some reports claim that the bond strength of a number of universal adhesives lags behind that of self-etching two-step adhesives. This has been observed with dentin in self-etch mode [10] and with enamel in etch-and-rinse and self-etch modes [11,12]. However, the versatility of universal adhesives has become more important in clinical practice [13], and further research is needed to apply universal adhesives more effectively.

According to Tian et al. [14], the chemical interaction of 10-MDP with dentin is essential for the preparation of strong compounds. From the interfacial research point of view, Inoue et al. [15] have shown that variations in dentin surface properties can be used to explain the chemical binding interactions between 10-MDP and dentin. The results on the chemical bonding interactions of universal adhesives on ground and etched dentin may have been inconsistent, although it has been previously demonstrated that universal adhesives have similar bonding performance regardless of the bonding methods used. In addition, further investigation of the different energetic properties of dentin surfaces treated with universal adhesives in the two different modes, in combination with bond fatigue resistance analysis, could provide a reason for the discrepancy between laboratory and actual results.

Adhesion of biomaterials to enamel and dentin may be compromised over time, causing bond breakdown and nanoleakage. Noncarious Class V clinical trials remain the gold standard for evaluating the efficacy of bonding, but they are also expensive, take time, and are labor intensive, and do not reveal the true reason for clinical failure. According to a study of modern adhesives, three-step etch-and-rinse adhesives remain the “gold standard” in terms of longevity. The clinical application procedure loses its adhesive power if it is simplified in any way. Only the two-step self-etching adhesives are close to the best and offer some further clinical advantages. When used in conjunction with hydrophobic resins, which may also contain fluoride and antimicrobial agents, solvent-free adhesives can seal resin-dentin surfaces. Compared to most 1- and 2-step adhesives, etch-and-rinse adhesives provide a stronger and more durable bond between resin and dentin. The strength of the resin-dentin bond can be improved by adding protease inhibitors to etchants or crosslinking agents to primers [10,16]. Phosphoric acid is used in etch-and-rinse systems to pretreat dental hard tissues before rinsing and subsequently applying an adhesive. Due to the presence of acidic monomers in self-etching adhesives, the tooth is simultaneously etched and primed, and pre-etching is not needed [17].

Scotchbond Universal Plus Adhesive (SB; 3M ESPE, Seefeld, Germany) and Clearfil Universal Bond (CU; Kuraray, Main, Germany) are widely tested universal adhesives. Their reported adhesive performances demonstrate their clinical suitability [18,19,20,21,22,23]. However, one of the major concerns of new universal adhesive systems was related to limited bond durability. Due to an increase in nanoleakage after aging, marginal discrepancies, secondary caries, and discoloration have been reported [24]. Nevertheless, when the clinical studies are completed, often a new version of the same material has already been made available on the market. One of the newest adhesives is G2-Bond Universal (G2-B; GC, Tokyo, Japan) and the manufacturer clams that its H-Technology decreases the risk of degradation and provides superior durability [25].

When universal adhesive is criticized, it is frequently noted that its thin film thickness allows oxygen to impede the polymerization of the adhesive layer for a significant amount of its depth. Suboptimal polymerization causes inadequate stabilization of the adhesive contact, which lowers the adhesive capacity of the layer to withstand stress. A universal adhesive’s ability to chemically attach to ceramics rich in glass could be compromised by the presence of integrated silane [26].

The present study aimed to compare the dentin bond durability of diverse universal adhesives after different storage and etching-modes. The following hypotheses were raised: 1) thermal cycling will not affect the shear bond strength of universal adhesives to dentin; 2) there will be no significant differences in dentin bond strength between different adhesives; and 3) etching with phosphoric acid will not increase the shear bond strength of the new G2-Bond Universal to dentin.

## 2. Materials and Methods

### 2.1. Sample Preparation

For the present study, we have collected forty (40) freshly extracted human permanent molars. The chosen molars were non-treated and caries free. Following removal of calculus deposits and soft tissues, molars were kept in 1% chloramine (KEFO; Sisak, Croatia) at 5 °C and used for the study in no later than one month from an extraction. The Ethics Committee of Zagreb University, School of Dental medicine approved this study (05-PA-30-XXVII-5/2021). Each molar was cut twice perpendicular to the long axis of the tooth, below and above the cement-enamel junction, obtaining a flat surface slab of dentin. The dentin slabs were thereupon cut through the center. Segmenting was performed with a precision cutter (Isomet 1000 Buehler; Lake Bluff, IL, USA) with diamond disk at 150 RPM and water cooling. All prepared specimens were placed with a bonding area upwards for testing in cylinder-shaped stainless-steel molds filled with cold-curing methacrylic resin. The casts were carefully removed after the resin’s curing and the specimens were divided into eight subgroups (*n* = 10), according to the adhesive, aging time, and adhesive procedure (Figure 1). Prior to adhesive procedures, the flat dentin surface was polished by a polishing machine (Le Cube, Presi; Grenoble, France) with P600 silicon carbide (SiC) abrasive paper, to ensure a uniformly even surface. As shown in Table 1, there were four sample groups, and in each group two subgroups. The subgroups contained ten (10) specimens, resulting in twenty (20) specimens per group. This sample size was determined according to a statistical analysis and performed by PASS NCSS, suggesting that twenty specimens per group allows statistical significant differences, when the groups are compared.

Materials used in the present study were three universal adhesives and a single bulk-fill resin composite (Table 2). The prepared specimens were copiously washed with distilled water, before carefully drying the dentin area. The dentin area was dried with a dry air spray, until there was no visible moisture. A polymer adhesive strip, with a circular cut 2.3 mm in diameter, was used to mark the bonding area. Each adhesive material was prepared according to manufacturers’ instructions, with G2-Bond Universal prepared with two different adhesive procedures (self-etch and total etch). The adhesive system was applied in a single layer, slowly air dried, but not immediately light cured. For polymerization we used Bluephase Style LED polymerization light (Ivoclar vivadent; Schaan, Liechtenstein) intensity of 1100 mW/cm^2^, which was measured using LED curing light radiometer Bluephase Meter II (Ivoclar Vicadent, Schaan, Liechtenstein). The specimens were placed into Ultradent Teflon mold (Ultra-dent Product, Inc., South Jordan, UT, USA) and composite cylinders (2.3 mm diameter × 3 mm height) were created by filling the mold with SDR flow+ (Dentsply Sirona; Charlotte, NC, USA) and clenching it. After the composite had set via light curing, the mold was disassembled and the specimens were stored in distilled water in an incubator (INEL, Zagreb, Croatia) at 37 °C for two months. Additionally, the second subgroup of the samples was subjected to a thermal cycling process for four days in distilled water. The set temperature for the baths was 5 °C and 55 °C. The storage time in each bath was 25 s, and the transfer time was 5 s [27].

### 2.2. Shear Bond Strength Testing

After the storage time of two months, half of the samples were loaded into a bond strength testing machine Ultra Tester (Ultradent Products, South Jordan, UT, USA) and tested into a macro shear mode. The machine was set to operate at a 1 mm/min crosshead speed until bond failure occurred. The second subgroup of samples was tested following the thermal cycling process. The shear bond strength values of the adhesives to dentin were regulated in accordance with ISO 29022 [28]. The fractured fragments were examined with a 3.6× optical loupe (Carl Zeiss Meditec AG, Oberkochen, Germany) and DinoLite microscope (DinoLite, Almere, The Netherlands) to determine the type of fracture, i.e., the cause of failure. If the fracture line is between the tooth and the composite cylinder, the fracture mode is classified as adhesive. The fracture mode is classified as mixed if the fracture line runs partially along the adhesive interface and penetrates one of the substrates, so we distinguish between the mixed fracture mode in the dentin or in the composite (depending on which substrate it covers). If more than 75% of the adhesive area involves either dentin or composite, the fracture mode is classified as cohesive. Surface morphology was examined by three examiners (E.K., A.I., and S.J.K) using different magnifications up to 200 magnification and photomicrographs of representative areas were taken. Figure 2 provides an illustration of the experimental research program.

### 2.3. Statistical Analysis

The statistical analysis was performed using two-way ANOVA with post hoc Tukey HSD test. The analyses were performed using IBM SPSS for Windows. The results were analyzed at a significance level of α = 0.05 at which the statistical power of the test was satisfactory (80%) to detect medium-sized effects (Cohen’s f = 0.25). 

## 3. Results

### 3.1. Shear Bond Strength (SBS)

Results revealed that SBS was significantly affected by the thermal cycling (TC) (*p* = 0.018) and different materials (*p* = 0.001), while the effect of interaction was not statistically relevant (*p* = 0.888). This means that the effect of thermal cycling on bond strength is similar for different materials. As shown in Figure 3, the lines connecting the estimated marginal means of MPa of thermal cycled samples and estimated marginal means of MPa of non-thermal cycled samples are approximately parallel. The highest difference between the samples subjected to thermal cycling and the ones which were not was measured in G2-BU in etch-and-rinse mode (Mean Difference = 4.7).

The statistically significant differences of the bond strengths between individual materials are presented in Table 3. In general, SB has the lowest MPa values (average values with and without TC) and is statistically significantly different from G2-BU (regardless of adhesive mode). MPa values of CU are between these two groups and no significant difference was observed compared to them (Table 2 and Table 3).

### 3.2. Type of Fractures

No significant differences were found between fracture types. All fractures, i.e., fracture lines, were located between the tooth and the composite cylinder, so the fracture mode was classified as adhesive (Figure 4 and Figure 5). None of the speciments were classified as mixed or cohesive.

## 4. Discussion

The performance of a restorative material in clinical practice is essential for material selection. Clinical research reveals the most reliable data regarding failure rates of a restorative treatment and its durability in the oral environment. However, it is very difficult for clinical research to determine a reason for a failed restoration and it is usually challenging to maintain a study‘s criteria throughout the time. This type of study also requires more time than a laboratory study, especially for testing the durability of a material. Considering the progressive evolution of the materials, often tested material is no longer in daily use by the time the study is completed [29]. This is the reason why adhesive systems are frequently chosen in accordance with the outcomes of laboratory testing, but it is important to remember that these tests are affected by a number of variables, including test specimen properties, specimen preparation, materials handling, specimen storage, test setup and test technique. Despite the fact that the shear test method is the most commonly used method for determining bond strength [30], several researchers believe it is of limited use in clinical performance assessment of dental adhesives, because the stress distribution is not as uniform as in a microtensile mode. There are counter-arguments to the tensile test in addition to the shear approach, such as the fact that restorations are hardly ever loaded in the tensile mode [31].

The adhesive system is one of the crucial factors in the success of the restoration. It makes the resin dental substrate interaction achievable [32]. Without the adhesive’s suitable mechanical properties choice of resin is unimportant because restoration is doomed to fail. Therefore, comparison of dentin bond strength of different new adhesive systems and testing their bond durability is meaningful. As a result, each of the loading tests given has strengths and weaknesses. Ultradent (Ultradent, Salt Lake City, UT, USA) developed the Ultradent jig to standardize the shear test technique [33]. This particular Ultradent jig makes contact with a broader specimen surface, encircling the specimen’s center and the composite material. By spreading the stress over a broader surface, the gadget can withstand higher load levels. Shear bond strength is assessed more accurately in this manner [34,35]. The ISO Technical Specification (TS) with the title “Testing the adhesion to tooth structure” (No. 11405, first edition 1994, second edition 2003, third edition 2015) lists a criterion titled “limiting of the bonding area is critical” that is routinely overlooked. 

Newly introduced and widely used universal adhesives were tested in this study. Clearfil SE Bond, introduced in 1991 and considered the gold standard in this category of self-etching adhesive systems that do not require phosphoric acid etching, is the industry leader in this field. The primers and adhesives in these systems often contain 10-methacryloyloxydecyl dihydrogen phosphate (10-MDP), which creates a durable and strong bond between the nanosheets and the calcium in the substrates [36]. After the expiration of Kuraray’s patent on 10-MDP, 3M Oral Care introduced Scotchbond Universal Adhesive in 2013, and numerous other manufacturers have since copied and improved this type of adhesive system. Scotchbond Universal Plus Adhesive is suitable for all etching processes, including direct and indirect bonding processes. It can also serve as a general primer for all restorative materials. To enhance bonding with dentin, many of these adhesives, including those previously mentioned, have used 2-hydroxyethyl methacrylate (HEMA) [37]. HEMA can easily penetrate demineralized substrate because it is hydrophilic, extremely dentin compatible, and water compatible. In contrast, its hydrophilicity makes it susceptible to hydrolysis and sorption, and it is known to cause allergic reactions [38]. For this reason, manufacturers have recently started to market HEMA-free adhesives. The durability of adhesives should be enhanced by eliminating HEMA in the primer and adhesive while reducing allergenicity. Recently, GC’s (Tokyo, Japan) two-step HEMA-free G2-Bond Universal, a novel adhesive that follows this strategy, was launched. G2-BOND Universal is newely developed 2-bottle adhesive with Dual-H technology which provides smoothly transitioning from hydrophilic to hydrophobic properties and enables advanced optimization of adhesion to tooth and composite. Due to the HEMA-free composition, the bonding layer is extremely hydrophobic, reducing the likelihood of water sorption, which reduces the risk of deterioration and results in excellent durability It also provides a robust bonding layer that prevents gap formation and microleakage [39].

The following hypotheses were tested: (i) thermal cycling will not affect shear bond strength of universal adhesives to dentin; (ii) there will be no significant differences in dentin bond strength between the tested materials; and (iii) etching with phosphoric acid will not increase the shear bond strength of new G2-BU to dentin. Results showed that shear bond strength was significantly affected by the thermal cycling and different adhesive systems use. SB had the lowest MPa values (average values with and without thermal-cycling) and is significantly different from G2-BU (regardless of adhesive mode), so the tested hypothesis that thermal cycling will not affect shear bond strength and that there will be no significant differences in dentin bond strength between the tested materials was therefore rejected. Furthermore, the new adhesive G2-BU had desirable shear bond strength to dentin compared to other universal adhesives, even after one year in the oral environment. The results also indicated that total-etching with phosphoric acid reduces shear bond strength of G2-BU to dentin, hence, the last hypothesis was rejected.

HEMA-free universal adhesive G2-BU was introduced to the market in the year 2021 as one of the newest adhesives technologies. The removal of HEMA in the primer and adhesive reportedly increases durability and reduces chances of allergic reactions. Although previous studies have tested initial bond strength and dentin bond strength after a day, there is no report of bond durability of G2-BU after a long period of time compared to representative adhesives [40]. Research conducted by Tsujimoto et al. has proven higher or equal dentin fatigue bond strength of G2-BU (fatigue resistance of G2-BU to dentin) in compression to SB Adhesive and CU. It is in correlation with this finding that the absence of HEMA increases the hydrophobicity of bonding agents and the strength of the cured layers. Hence, G2-BU has a higher hydrophobic effect than CU and even greater than SB [41]. The present study supports this claim, because the results showed a significantly higher G2-BU dentin shear bond strength compared to SB. The values of CU lacked statistical significance in comparison to other tested materials. 

A wide range of outcomes are possible when interpreting the data of the bond strength test [42]. On the other side, the main reason for variability is the brittle nature of materials such as dental adhesives and composites [43]. The maximum stress that brittle materials can bear differs surprisingly from sample to sample, even when a collection of ostensibly comparable samples is tested under the same conditions [44]. Brittle materials’ assessed strength is predicated on the possibility of a critical defect emerging in materials’ structure since their strength is previously determined by the defects or imperfections already present in the samples [45]. The number of tests conducted, the mean strength, and the standard deviation [46,47] are the three data elements most frequently used to depict bond strength testing results, which are composed of measurements done on numerous specimens that appear to be identical [48]. This concept deems the mean value as the “true value” and suggests that variations in test methods or specimen preparation are to blame for data scattering around the real value. 

In order to avoid waiting a long period of time to pass to determine a durability of the material, there are several methods for aging samples in vitro. In the oral environment, restorations are exposed to chemical, physical and mechanical processes; thus, there are mechanical, biological, chemical and physical aging factors. Given that determining the bond strength was the aim of this study and that the biggest deficiency of universal adhesives is its tendency for physical degradation, we decided to use thermocycling as an aging process. Although there is no standardized protocol for laboratory aging of restorative material, studies have shown that thermal cycling (5/55 °C/1 min) for four days is one of the most effective aging methods. Thus, it was the method used in the present study [27,49]. Being in vitro, the study assessed bond strength of the universal adhesives under controlled laboratory conditions. The long-term bonding performance in an oral environment, influenced by various factors such as oral hygiene, saliva, oral microflora, occlusal forces and patient behavior, was not taken into account. Results suggested that shear bond strength of tested materials was significantly affected by the thermal cycling, but the effect of thermal cycling on bond strength was similar for all the materials. On the other hand, G2-BU post-thermal cycling exhibited significantly higher values of dentin bond strength in self-etch mode compared to etch-and-rise mode. A similar result was found in past studies about universal adhesives, where Hanabusa et al. and Leite et al. made an observation that the resulting adhesive interface in the etch-and-rinse approach was ultra-structurally more exposed to biodegradation compared to the self-etch approach, resulting in the decrease in bond strength post aging [50,51]. 

Together, these findings show that depending on the type of adhesive and the manner of application for the same adhesive, the bonding process for universal adhesives can differ dramatically. A deeper understanding of these concepts could significantly increase the bonding ability of universal adhesives, making this a crucial subject for additional research. Because the etch-and-rinse method did not produce stronger enamel bonds than the self-etch method and has less effect on dentin’s fatigue resistance, a universal adhesive applied in this method would have lower fatigue resistance. Clinical trials, however, revealed that when a universal bonding method was applied, there was no statistically significant difference between different dentin preparation [11,12]. Considering all the results, G2-BU in the self-etch approach is the most efficient way to secure high dentin bond strength, compared to other representative adhesives.

## 5. Conclusions

With the limitation of not simulatining the aging phenomenon in the oral cavity in this in vitro study, this study implies that the HEMA-free universal adhesive G2-Bond Universal showed higher or equal dentin bond strength than representative adhesive and the most stable dentin bond, particularly when applied in the self-etch mode. As a result, additional etching can weaken the bond. Further experiments are needed, such as comparisons of different universal adhesives, as well as conventional and bulk composites in different application approaches and analyzing the bond strength of specimens over different time periods.

## Figures and Tables

**Figure 1 materials-16-01530-f001:**
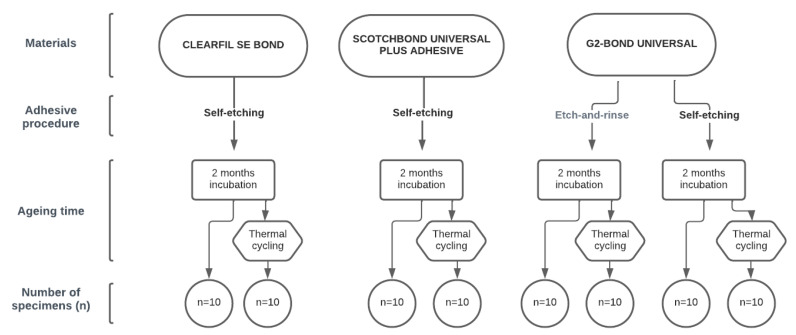
Samples’ groups.

**Figure 2 materials-16-01530-f002:**
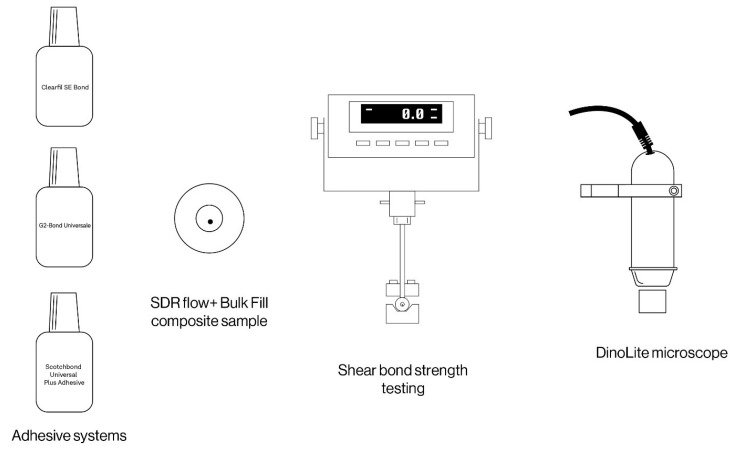
Shear bond strength testing and surface microanalysis.

**Figure 3 materials-16-01530-f003:**
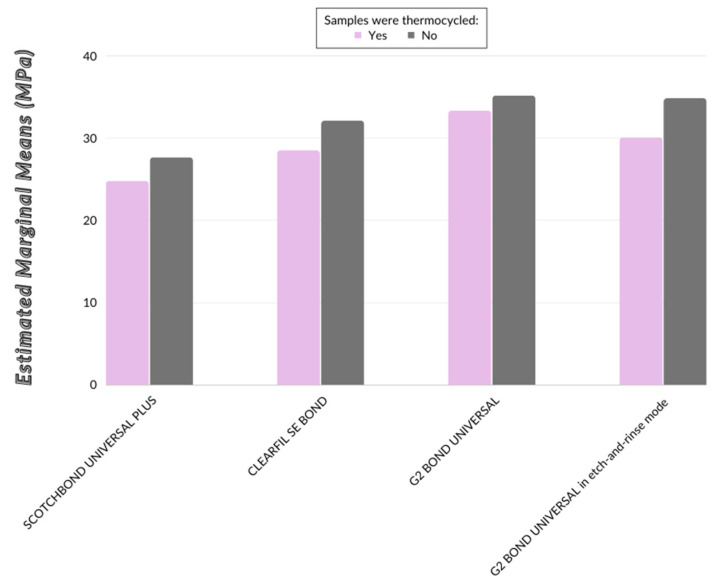
Estimated marginal means of SBS of the tested adhesives.

**Figure 4 materials-16-01530-f004:**
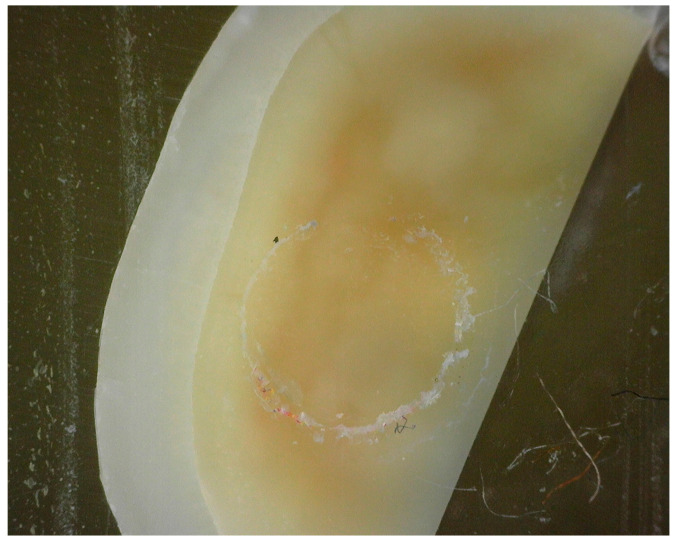
Adhesive fracture between the composite sample and dentin surface.

**Figure 5 materials-16-01530-f005:**
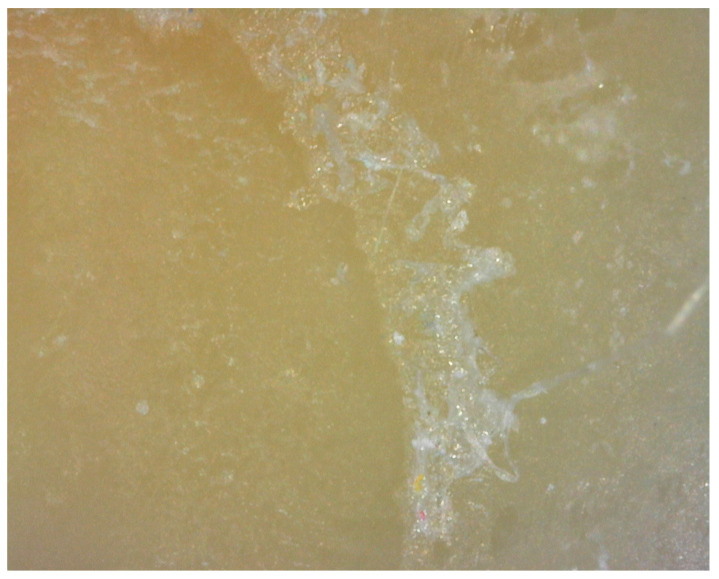
Adhesive fracture between the composite sample and dentin surface.

**Table 1 materials-16-01530-t001:** The study’s materials.

Material	Type of Material	Main Components	Manufacturer
Clearfil SE Bond (CU)	Two-step Universal Adhesive	Primer: HEMA^*^, 10-MDP ^*^, initiators, water Adhesive: 2-HEMA ^*^, 10-MDP ^*^, Bis-GMA ^*^, photoinitiator	Kuraray; Main, Germany
G2-Bond Universale (G2-BU)	Two-step Universal Adhesive	Primer: 10-MDTP ^*^, 10-MDP ^*^, 4-MET ^*^, aceton, water, initiators, fillers, water Adhesive: Bis-GMA ^*^, dimethacrylate monomer, filler, photoinitiator	GC; Tokyo, Japan
Scotchbond Universal Plus Adhesive (SB)	One-step Universal Adhesive	Adhesive: 2-HEMA ^*^, 10-MDP ^*^, Bis-GMA ^*^, ethanol, photoinitiator, fillers, water	3M; Seefeld, Germany
Ultra-Etch	Pre-etching agent	35% phosphoric acid	Ultradent Products; South Jordan, UT, USA
SDR flow+ Bulk Fill Flowable	Resin composite	Proprietary modified urethane dimethacrylate resin, TEGDMA ^*^, photoinitiator, fluorescent agent, fluoro-silicate glass, surface treated fume silicas, fluoride, pigments, titanium dioxide	Dentsply Sirona; Charlotte, NC, USA

^*^ According to the manufacturers’ information. HEMA: 2-hydroxyethyl methacrylate; BisGMA: Bisphenol-A glycidyl methacrylate; 10-MDP: 10-methacryloyloxydecyl dihydrogenphosphate; 4-MET: 4-methacryloxyethyl trimellitic acid; MDTP: methacryloyloxydecyl dihydrogen thiophosphate; TEGDMA: triethylene glycol dimethacrylate.

**Table 2 materials-16-01530-t002:** The effect of multiple comparisons between studied groups.

(I) Materials	(J) Materials	Mean Difference (I-J)	Std. Error	Sig.	95% Confidence Interval	
					Lower Bound	Upper Bound
SB	CU	−4.085	1.909	0.151	−9.106	0.936
	G2-BU	−8.030 *	1.909	0.000	−13.051	−3.008
	G2-BU in etch-and-rinse mode	−6.245 *	1.909	0.009	−11.266	−1.223
CU	SB	4.085	1.909	0.151	−0.936	9.106
	G2-BU	−3.945	1.909	0.174	−8.966	1.076
	G2-BU in etch-and-rinse mode	−2.160	1.909	0.671	−7.181	2.861
G2-BU	SB	8.030 *	1.909	0.000	3.008	13.051
	CU	3.945	1.909	0.174	−1.076	8.966
	G2-BUin etch-and-rinse mode	1.785	1.909	0.786	−3.236	6.806
G2-BU in etch-and-rinse mode	SB	6.245 *	1.909	0.009	1.223	11.266
	CU	2.160	1.909	0.671	−2.861	7.181
	G2-BU	−1.785	1.909	0.786	−6.806	3.236
Based on observed means. The error term is Mean Square (Error) = 36.451.						

* The mean difference is significant at the 0.05 level.

**Table 3 materials-16-01530-t003:** Mean ± standard deviation of SBS of the tested adhesive application mode and aging combinations.

		**2 Months Incubation and Thermal Cycling**			**2 Months Incubation without Thermal Cycling**			** *p* **
		**MPa**			**MPa**			
		**Mean**	**Standard Deviation**	**Count**	**Mean**	**Standard Deviation**	**Count**	
Material	SB	24.78	5.98	10	27.64	5.19	10	0.268
	CU	28.49	6.35	10	32.10	5.69	10	0.197
	G2-BU	33.33	6.27	10	35.15	6.58	10	0.535
	G2-BU in etch-and-rinse mode	30.07	5.81	10	34.84	6.31	10	0.096
	P among groups overall	0.028			0.028			

The error term is Mean Square (Error) = 36.451. Confidence interval = 95%

## Data Availability

The data that support the findings of this study are available from the corresponding author upon request.

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
