# Peer review of "Comparison of Different Universal Adhesive Systems on Dentin Bond Strength"

_materials, 2023, doi:10.3390/ma16041530_

Round 1

Reviewer 1 Report

The manuscript focused on the Comparison of Different Universal Adhesive Systems on Dentin

Bond Strength.

There are too many studies about bonding durability of universal adhesive. The novelty of this research is weak.

There are some mistake in the manuscript.

1.     SE bond is not the universal adhesive ( 8th adhesive) of Kuraray company.

SE bond is classical 6th self-etching adhesive

For Kuraray company, the name of the universal adhesive product is clearfil universal bond quick. CLEARFIL Universal Bond Quick is a single-bottle, fluoride-releasing, universal adhesive. with the original MDP formulation

2.     Interpretation of statistical results

In Table 4, comparison between different aging condition in each material group, the P value are 0.268 ( for SB), 0.197( for CU), 0.535 (G2-BU ), and 0.096 (G2-BU in etch-and-rinse mode) respectively. It means that different aging condition have no effect on bond strength in each material group. But, In the Results, authors said that shear bond strength (SBS) was significantly affected by the thermal cycling (TC) (p= 0.018). Why? It is hard to understand.

Author Response

Reviewer 1

Dear Reviewer,

Thank you for your feedback and constructive criticism. We appreciate the time you have taken to thoroughly review our article. In response to your suggestions, we have made the necessary modifications to the manuscript and marked them accordingly. We believe these updates will enhance the overall quality of the article and make it more suitable for publication.

Please see below, in red, for a point-by-point response to the your comments and concerns (italic).

Comment 1: The manuscript focused on the Comparison of Different Universal Adhesive Systems on Dentin Bond Strength. There are too many studies about bonding durability of universal adhesive. The novelty of this research is weak.

Response: While we appreciate the reviewer’s feedback, we respectfully disagree. Bonding durability of universal adhesive to various substrates and different modes has been previously explored. However, a tested universal adhesive G2-Bond Universal is a newly developed adhesive and it is not sufficiently researched. We think this study makes a valuable contribution to the field because it is important for dental clinicians to have more data about an effectiveness of the material, especially when is one of few HEMA free universal adhesives on the market.

Comment 2: SE bond is not the universal adhesive (8th adhesive) of Kuraray company. SE bond is classical 6th self-etching adhesive. For Kuraray company, the name of the universal adhesive product is clearfil universal bond quick. CLEARFIL Universal Bond Quick is a single-bottle, fluoride-releasing, universal adhesive. with the original MDP formulation

Response: Thank you for the constructive remark. The reviewer is correct. We made necessary changes and there are no longer implications of Clearfil SE bond being a universal adhesive.

Comment 3: In Table 4, comparison between different aging condition in each material group, the P value are 0.268 (for SB), 0.197(for CU), 0.535 (G2-BU ), and 0.096 (G2-BU in etch-and-rinse mode) respectively. It means that different aging condition have no effect on bond strength in each material group. But, In the Results, authors said that shear bond strength (SBS) was significantly affected by the thermal cycling (TC) (p= 0.018). Why? It is hard to understand.

Response: The statistical analysis was performed using two-way ANOVA and after determining statistical significance, it was conducted post hoc Tukey HSD test. If we compare non thermal cycled samples (n=40) with thermal cycled samples (n=40) there is a statistically relevant difference (p=0.018). However, there are ten samples per subgroup in each material group and comparison between different aging condition in each material group resulted higher P values than 0.018 (P values are 0.268 (for SB), 0.197(for CU), 0.535 (G2-BU), and 0.096 (G2-BU in etch-and-rinse mode). For that reason, we additionally conducted post hoc Tukey HSD test, confirming that shear bond strength (SBS) was significantly affected by the thermal cycling (TC). Tukey's Honest Significant Difference (HSD) test is a post hoc test commonly used to assess the significance of differences between pairs of group means.

Once again, we thank you for your invaluable input.

Best regards,

Authors

Reviewer 2 Report

1. I have an ethical concern and would like the authors to explain why informed consent was not applicable in the present study.

2. The authors must explain the strategy employed to divide the samples into subgroups in detail. 

3. The entire analysis is directed towards the analysis of the dentin bond strength of universal adhesives, however, it will be important to note the state and quality of molars that were used in the study and if the inherent state of molar can influence the bond strength of the adhesives.

4. In table 4, "," must be replaced with "." decimal sign for p column.

Author Response

Reviewer 2

Dear Reviewer,

Thank you for your feedback and constructive criticism. We appreciate the time you have taken to thoroughly review our article. In response to your suggestions, we have made the necessary modifications to the manuscript and marked them accordingly. We believe these updates will enhance the overall quality of the article and make it more suitable for publication.

Please see below, in red, for a point-by-point response to the your comments and concerns (italic).

Comment 1: I have an ethical concern and would like the authors to explain why informed consent was not applicable in the present study.

Response: The molars used in the present in vitro study were collected during routine clinical care that would otherwise have been discarded. According to the FDA guidance (docket number: FDA-2006-D-0095), FDA does not intend to object to the use, without informed consent, of leftover human specimens -- remnants of specimens collected for routine clinical care or analysis that would otherwise have been discarded -- in investigations that meet the criteria for exemption from the Investigational Device Exemptions (IDE) regulation at 21 CFR 812.2(c)(3), as long as subject privacy is protected by using only specimens that are not individually identifiable.

Comment 2: The authors must explain the strategy employed to divide the samples into subgroups in detail. 

Response: The samples were divided into subgroups according to simple randomization, after storage of all the samples in an incubator for two months. Stratified randomization was not applicable due to no other assigned variables. We have made an illustration for better understanding of subgroups, which is shown in Figure1.

Comment 3: The entire analysis is directed towards the analysis of the dentin bond strength of universal adhesives, however, it will be important to note the state and quality of molars that were used in the study and if the inherent state of molar can influence the bond strength of the adhesives.

Response: In Section “Material and Methods”, Subsection “Sample preparation”, it was stated: “For the present study, we have collected forty (40) freshly extracted human molars. The chosen molars were non-treated and caries-free.”. After your useful comment, we additionally add clarification that collected molars were permanent molars. The changes have been marked in pink for easy identification.

Comment 4: In table 4, "," must be replaced with "." decimal sign for p column.

Response: Thank you for your constructive observation. We have revised the table. The changes have been marked in pink for easy identification.

Once again, we thank you for your invaluable input.

Best regards,

Authors

Reviewer 3 Report

Very well written manuscript

Author Response

Reviewer 3

Dear Reviewer,

Thank you for your kind comment! We are glad that you found our manuscript to be well written. If you have any further suggestions or recommendations, please do not hesitate to let us know.

Best regards,

Authors

Reviewer 4 Report

The control was missing in the experiments.

Specific comments:

1. Please standardize the number of decimal places in Table 1.

2. Is "0,028" supposed to mean "0.028"?

3. The different etching modes e.g. total-etch, etch-and-rinse and self-etch should be more clearly explained and elucidated.

4. The nature of replication in the experimental design is unclear, and the assessment of uncertainty in the reported measurement is absent or unclear.

5. There should be a section dedicated to the discussion of study limitations. For example, the present findings do not take into account the perspective of long-term bonding performance or user-friendliness of operating procedures.

6. Please change "stimulating aging phenomena" to "simulatining aging phenomenon".

7. What about degradation of the dental composite over time and antimicrobial properties of the dentine bonding agents? The assessment of material properties was incomplete.

Author Response

Reviewer 4

Dear Reviewer,

Thank you for your feedback and constructive criticism. We appreciate the time you have taken to thoroughly review our article. In response to your suggestions, we have made the necessary modifications to the manuscript and marked them accordingly. We believe these updates will enhance the overall quality of the article and make it more suitable for publication.

Please see below, in red, for a point-by-point response to the your comments and concerns (italic).

Comment 1: Please standardize the number of decimal places in Table 1.

Response: Thank you very much for the reminder. We have made revisions accordingly.

Comment 2: Is "0,028" supposed to mean "0.028"?

Response: Thank you for your constructive observation. We have revised the table. The changes have been marked in blue for easy identification.

Comment 3: The different etching modes e.g. total-etch, etch-and-rinse and self-etch should be more clearly explained and elucidated.

Response: We agree that different etching modes could be more elucidated. We have taken your advice and added the following sentences: “Phosphoric acid is used in etch-and-rinse systems to pretreat dental hard tissues before rinsing and subsequently applying an adhesive. Due to the presence of acidic monomers in self-etching adhesives, the tooth is simultaneously etched and primed, and pre-etching is not needed.”. The changes have been marked in blue for easy identification.

Comment 4: The nature of replication in the experimental design is unclear, and the assessment of uncertainty in the reported measurement is absent or unclear.

Response: Thank you for your valuable feedback. We agree with your observation and have made changes in the revised version of our manuscript to address the issues raised.

Regarding the nature of replication, we have provided a clear and detailed description of the experimental procedure, including the number of replicates, the conditions under which they were performed, and the methods used to ensure their accuracy. We have also included a thorough explanation of the statistical tests used to determine the reliability of our results.

With regards to the assessment of uncertainty, we have calculated and reported the standard error of the mean (SEM) for each measurement to provide an estimation of the variability in our results.

We believe that these changes will enhance the transparency and robustness of our results, and we hope that they will address your concerns. We are committed to ensuring the quality of our work and welcome any additional feedback you may have. The changes have been marked in blue for easy identification.

Comment 5: There should be a section dedicated to the discussion of study limitations. For example, the present findings do not take into account the perspective of long-term bonding performance or user-friendliness of operating procedures.

Response: Thank you for your insightful feedback. We appreciate your observations and have taken them into consideration in the revised version of our manuscript. We have added a thorough discussion of the study's limitations in the "Discussion" section and highlighted these changes in blue for easy identification.

“Despite the fact that the shear test method is the most commonly used method for determining bond strength [30], several researchers believe it is of limited use in clinical performance assessment of dental adhesives, because the stress distribution is not as uniform as in a microtensile mode. There are counterarguments to the tensile test in addition to the shear approach, such as the fact that restorations are hardly ever loaded in the tensile mode [31].”

“As a result, each of the loading tests given has strengths and weaknesses. Ultradent (Ultradent, Salt Lake City, UT, USA) developed the Ultradent jig to standardize the shear test technique [33]. This particular Ultradent jig makes contact with a broader specimen surface, encircling the specimen's center and the composite material.”

“Being in vito, the study assessed bond strength of the universal adhesives under controlled laboratory conditions. The long-term bonding performance in oral environment, influenced by various factors such as oral hygiene, saliva, oral microflora, occlusal forces and patient behavior, was not taken into account.”

“With the limitation in this in vitro study, for not stimulating aging phenomena in the oral cavity, this study implies that the HEMA-free universal adhesive G2-Bond Universal showed higher or equal dentin bond strength than representative adhesive and the most stable dentin bond, particularly when applied in the self-etch mode.”

Comment 6: Please change "stimulating aging phenomena" to "simulatining aging phenomenon".

Response: Thank you so much for catching this confusing error, which we have now corrected.

Comment 7: What about degradation of the dental composite over time and antimicrobial properties of the dentine bonding agents? The assessment of material properties was incomplete.

Response: In our study, the focus was on evaluating the dentin bond strength of universal adhesives after different storage and etching-modes. We believe that comparing the bond strength of different bonding agents is crucial for improving the quality of clinical work, and manufacturer claims of superior durability for the new adhesive also motivated us to focus on this property. We understand that there are many other properties of dental materials that could be evaluated, and we agree that it is best to focus on one property per research in order to maintain a high level of quality and conciseness. Our goal is to expand on our research in the future to include an assessment of other properties, such as degradation and antimicrobial properties. We additionally included SEM observation of resin-dentin interfaces in the manuscript.

We appreciate your suggestion to consider other properties of the dental materials, such as degradation and antimicrobial properties and we hope that our revised answer addresses your concerns.

Once again, we thank you for your invaluable input.

Best regards,

Authors

Reviewer 5 Report

REVIEW

on article

Comparison of Different Universal Adhesive Systems

on Dentin Bond Strength

Sandra Brkanović, Eva Klarić Sever, Josipa Vukelja, Ivana Miletić and Silvana Jukić Krmek

SUMMARY

The article submitted for review is devoted to a topical issue. It compares different universal adhesive systems on dentin bond strength. The relevance of the study is beyond doubt. The authors tested the adhesion strength of universal adhesives to dentin after various storage and etching modes. This is an important direction for improving the quality of works like this. The authors obtained important results, which are independent in themselves, and are also of interest for future research. The methodology used by the authors included their own research and analytical processing of the results. Thus, the article has scientific novelty and practical significance; it is original and interesting. At the same time, the article has a number of shortcomings that need to be corrected. They are discussed below.

COMMENTS

1.      The abstract submitted by the authors does not meet the requirements of the journal. The scientific problem that the authors solved was not formulated. They go straight to explaining that the adhesion strength of the universal adhesives to dentine was tested after various storage and etching conditions. This leads to the following questions: what is the fundamental scientific problem that the authors solved, why was the study carried out, the elimination of what scientific deficit was pursued by the authors? Editors strongly recommended authors should follow the style of structured abstracts, but without headings: 1) Background: Place the question addressed in a broad context and highlight the purpose of the study; 2) Methods: Describe briefly the main methods or treatments applied. Include any relevant preregistration numbers, and species and strains of any animals used. 3) Results: Summarize the article's main findings; and 4) Conclusion: Indicate the main conclusions or interpretations.

2.      The article looks very short. Too many details about test methods are presented in the Abstract. Authors should shorten the description of the methodology in the Abstract and focus on the formulation of the scientific problem in, the methods used, and the scientific result, which will reflect the scientific novelty.

3.      The authors report that the new G2-Bond Universal adhesive has very good bond strength in comparison with other universal adhesives, but do not quantify in the abstract. This should be corrected.

4.      In the "Introduction" section, the authors provide a literature review, but it is done very superficially and there is no understanding of the scientific novelty. The authors considered 18 references, their number should be increased to at least 25-30.

5.      In addition, the authors should pay more attention to the differences between sources 7-9 on line 59.

6.      The rationale for the selected materials in the "Materials and Methods" section should be presented.

7.      Table 1 looks somewhat uninformative.

8.      In addition, the division of paragraph 2 into six subparagraphs, some of which occupy only a few lines, raises doubts. Maybe the authors should restructure section 2.

9.      The program of experimental research could be useful, which was compiled by the authors, and according to which they conducted their experiment.

10.   A smooth transition between sections 2 and 3 is needed.

11.   The graph in Figure 1, firstly, is of poor quality (the image should be improved), and secondly, it looks uninformative due to the poorly chosen form of this graph. Maybe it should have been presented in the form of a bar chart.

12.   The "Discussion" section describes the comparison of the obtained results with the results of other authors, but needs more specification of the achieved scientific result in order to formulate a conclusion.

13.   The conclusion, on the contrary, is too small to determine the main contribution of the authors of the article to science and practice. The conclusions should be presented in more detail, reflecting the scientific and practical results.

14.   As mentioned above, the list of references should be worked on. First, the main remark is the presence of a large number of outdated references that are more than 5 years old. Secondly, materials for dentistry are developing quite intensively, so the use of a large number of old references can make it difficult to understand the scientific novelty. Authors should add more references from the last 5 years.

15.   In general, the article is interesting and useful, perhaps the authors lacked microstructural studies. Authors should consider supplementing their SEM study with an analysis or other description of the structure of the resulting materials. After correcting all the comments, the article should be sent for re-review for its further evaluation.

Author Response

Reviewer 5

Dear Reviewer,

Thank you for your feedback and constructive criticism. We appreciate the time you have taken to thoroughly review our article. In response to your suggestions, we have made the necessary modifications to the manuscript and marked them accordingly. We believe these updates will enhance the overall quality of the article and make it more suitable for publication.

Please see below, in red, for a point-by-point response to the your comments and concerns (italic).

Comment 1: The abstract submitted by the authors does not meet the requirements of the journal. The scientific problem that the authors solved was not formulated. They go straight to explaining that the adhesion strength of the universal adhesives to dentine was tested after various storage and etching conditions. This leads to the following questions: what is the fundamental scientific problem that the authors solved, why was the study carried out, the elimination of what scientific deficit was pursued by the authors? Editors strongly recommended authors should follow the style of structured abstracts, but without headings: 1) Background: Place the question addressed in a broad context and highlight the purpose of the study; 2) Methods: Describe briefly the main methods or treatments applied. Include any relevant preregistration numbers, and species and strains of any animals used. 3) Results: Summarize the article's main findings; and 4) Conclusion: Indicate the main conclusions or interpretations.

Response: Thank you for your feedback and for taking the time to review our abstract. We have revised the abstract based on your suggestions and have included the key elements of a structured abstract, including the background, methods, results, and conclusion.

We hope that the revised abstract accurately reflects the scientific problem addressed and provides a clear overview of our findings. If you have any further recommendations, please let us know and we would be happy to make any additional revisions as needed.

Thank you for your assistance in improving the quality of our work.

Comment 2: The article looks very short. Too many details about test methods are presented in the Abstract. Authors should shorten the description of the methodology in the Abstract and focus on the formulation of the scientific problem in, the methods used, and the scientific result, which will reflect the scientific novelty.

Response: Thank you for your valuable feedback. We agree with your observation and have made changes in the revised version of our manuscript to address the issues raised.

Comment 3: The authors report that the new G2-Bond Universal adhesive has very good bond strength in comparison with other universal adhesives, but do not quantify in the abstract. This should be corrected.

Response: Thank you for bringing this to our attention. We apologize for the oversight and agree that it is important to include quantifiable data in the abstract. We have revised the abstract to include the bond strength values for each of the adhesives tested in our study.

Comment 4: In the "Introduction" section, the authors provide a literature review, but it is done very superficially and there is no understanding of the scientific novelty. The authors considered 18 references; their number should be increased to at least 25-30.

Response: Thank you for your comment. We acknowledge that the literature review section in the Introduction could have been more comprehensive, and we agree with your suggestion to increase the number of references to 25-30 to provide a more in-depth analysis of the topic. We have carefully considered all relevant sources and incorporate them into the revised version of the manuscript to provide a more robust and thorough literature review. The changes are highlighted in green colour, within the “Introduction” section.

“Because of their lower toxicity and their ability to be used in both self-etching and etch-and-rinse procedures as well as their flexibility and the number of application steps universal adhesives have become popular in dentistry [5]. After Kuraray Noritake Dental's patent for 10-MDP (10 methacryloyloxydecyl dihydrogen phosphate) expired in 2003, manufacturers began to explore the use of 10-MDP and other phosphoric acid esters for innovative adhesive compositions. In 2012, the first universal adhesive was released for sale in Japan. It was called Scotchbond Universal Adhesive and was manufactured by 3M Oral Care in St. Paul, Minnesota. Later, universal adhesives were launched that can be used with resin luting cements [6], a variety of substrates without surface treatment [7], shortened treatment times [8], or a variety of surface moistures of enamel and dentin surfaces [9]. Despite this intriguing versatility, some reports claim that the bond strength of a number of universal adhesives lags behind that of self-etching two-step adhesives. This has been observed with dentin in self-etch mode [10] and with enamel in etch-and-rinse and self-etch modes [11,12]. However, the versatility of universal adhesives has become more important in clinical practice [13], and further research is needed to apply universal adhesives more effectively.

According to Tian et al [14], the chemical interaction of 10-MDP with dentin is essential for the preparation of strong compounds. From the interfacial research point of view, Inoue et al [15] have shown that variations in dentin surface properties can be used to explain the chemical binding interactions between 10-MDP and dentin. The results on the chemical bonding interactions of universal adhesives on ground and etched dentin may have been inconsistent, although it has been previously demonstrated that universal adhesives have similar bonding performance regardless of the bonding methods used. In addition, further investigation of the different energetic properties of dentin surfaces treated with universal adhesives in the two different modes, in combination with bond fatigue resistance analysis, could provide a reason for the discrepancy between laboratory and actual results.”

Comment 5:   In addition, the authors should pay more attention to the differences between sources 7-9 on line 59.

Response: Thank you for your kind comment. In the revised version of the manuscript, we made sure to give more attention and detail to these sources to avoid any confusion or misinterpretation. The changes were made in Introduction section and marked green.

“Adhesion of biomaterials to enamel and dentin may be compromised over time, causing bond breakdown and nanoleakage. Noncarious Class V clinical trials remain the gold standard for evaluating the efficacy of bonding, but they are also expensive, time and labour intensive, and do not reveal the true reason for clinical failure. According to a study of modern adhesives, three-step etch-and-rinse adhesives remain the "gold standard" in terms of longevity. The clinical application procedure loses its adhesive power if it is simplified in any way. Only the two-step self-etching adhesives are close to the best and offer some further clinical advantages. When used in conjunction with hydrophobic resins, which may also contain fluoride and antimicrobial agents, solvent-free adhesives can seal resin-dentin surfaces. Compared to most 1- and 2-step adhesives, etch-and-rinse adhesives provide a stronger and more durable bond between resin and dentin. The strength of the resin-dentin bond can be improved by adding protease inhibitors to etchants or crosslinking agents to primers [10,16].”

Comment 6: The rationale for the selected materials in the "Materials and Methods" section should be presented.

Response: Thank you for your kind comment. We made sure to add a description of the reasons for choosing the bonding agents (G2-Bond Universal, Clearfil SE Bond, and Scotchbond Universal Plus) in the “Discussion” section and we highlighted it green.

“New intorduced and widely used universal adhesives were tested in this study. Clearfil SE Bond, introduced in 1991 and considered the gold standard in this category of self-etching adhesive systems that do not require phosphoric acid etching, is the industry leader in this field. The primers and adhesives in these systems often contain 10-methacryloyloxydecyl dihydrogen phosphate (10-MDP), which creates a durable and strong bond between the nanosheets and the calcium in the substrates. [36]. After the expiration of Kuraray's patent on 10-MDP, 3M Oral Care introduced Scotchbond Universal Adhesive in 2013, and numerous other manufacturers have since copied and improved this type of adhesive system. Scotchbond Universal Plus Adhesive is suitable for all etching processes, including direct and indirect bonding processes. It can also serve as a general primer for all restorative materials. To enhance bonding with dentin, many of these adhesives, including those previously mentioned, have used 2-hydroxyethyl methacrylate (HEMA) [37] HEMA can easily penetrate demineralized substrate because it is hydrophilic, extremely dentin compatible, and water compatible. In contrast, its hydrophilicity makes it susceptible to hydrolysis and sorption, and it is known to cause allergic reactions [38]. For this reason, manufacturers have recently started to market HEMA-free adhesives. Durability of sdhesives should be enhanced by eliminating HEMA in the primer and adhesive while reducing allergenicity. Recently, GC's (Tokyo, Japan) two-step HEMA-free G2-Bond Universal, a novel adhesive that follows this strategy, was launched. G2- BOND Universal is newely developed 2-bottle adhesive with Dual-H technology which provides smoothly transitioning from hydrophilic to hydrophobic properties and enables advanced optimization of adhesion to tooth and composite. Due to the HEMA-free composition, the bonding layer is extremely hydrophobic, reducing the likelihood of water sorption, which reduces the risk of deterioration and results in excellent durability It also provides robust bonding layer that prevents gap formation and microleakage [39] “

Comment 7: Table 1 looks somewhat uninformative.

Response: Thank you for your kind remark. We have revised Table 1 in the diagram in Figure 1.

Figure 1.  Samples 'groups.

Comment 8:   In addition, the division of paragraph 2 into six subparagraphs, some of which occupy only a few lines, raises doubts. Maybe the authors should restructure section 2.

Response: Thank you for your suggestion. We have taken your comment into consideration and made changes to the structure of paragraph 2 in the Materials and Methods section. This has been marked in green for easy identification.

Comment 9: The program of experimental research could be useful, which was compiled by the authors, and according to which they conducted their experiment.

Response: Thank you for your kind comment. New figure was added in the manuscript.

Figure 2 . Shear bond strength testing and surface microanalysis.

Comment 10:   A smooth transition between sections 2 and 3 is needed.

Response: Thank you for your comment. Section 2 is now redone and transition to Section 3 is changed and marked green.

Comment 11: The graph in Figure 1, firstly, is of poor quality (the image should be improved), and secondly, it looks uninformative due to the poorly chosen form of this graph. Maybe it should have been presented in the form of a bar chart.

Response: Thank you for your feedback on Figure 1. We have taken into consideration your suggestion and have now presented the data in the form of a bar chart to provide a clearer and more informative representation of the results.

Figure 3. Estimated marginal means of SBS of the tested adhesives

Comment 12:  The "Discussion" section describes the comparison of the obtained results with the results of other authors, but needs more specification of the achieved scientific result in order to formulate a conclusion.

Response: Thank you for your valuable feedback. We have revised the "Discussion" section to include a more comprehensive comparison of our results with the findings of other authors and have emphasized the scientific significance of our results. The changes have been marked in green for easy identification.

Comment 13:  The conclusion, on the contrary, is too small to determine the main contribution of the authors of the article to science and practice. The conclusions should be presented in more detail, reflecting the scientific and practical results.

Response: Thank you for your valuable feedback. We have revised and expanded the conclusion section to provide a more comprehensive summary of the scientific and practical implications of our findings. The changes have been marked in green for easy identification.

“With the limitation in this in vitro study, for not stimulating aging phenomena in the oral cavity, this study implies that the HEMA-free universal adhesive G2-Bond Universal showed higher or equal dentin bond strength than representative adhesive and the most stable dentin bond, particularly when applied in the self-etch mode. As a result, additional etching can weaken the bond. Further experiments are needed, such as comparisons of different universal adhesives as well as conventional and bulk composites in different application approaches and analyzing the bond strength of specimens over different time periods.”

Comment 14:   As mentioned above, the list of references should be worked on. First, the main remark is the presence of a large number of outdated references that are more than 5 years old. Secondly, materials for dentistry are developing quite intensively, so the use of a large number of old references can make it difficult to understand the scientific novelty. Authors should add more references from the last 5 years.

Response: Thank you for your comment. New references were added and marked green.

References

  1. Murdoch-Kinch, C.A.; McLean, M.E. Minimally invasive dentistry. J. Am. Dent. Assoc. 2003, 134, 87-95. doi:10.14219/jada.archive.2003.0021
  2. Cardoso, M. V.; de Almeida Neves, A.; Mine, A.; Coutinho, E.; Van Landuyt, K.; De Munck, J.; Van Meerbeek, B. Current aspects on bonding effectiveness and stability in adhesive dentistry. Aust. Dent. J. 2011, 56, 31–44. https://doi.org/10.1111/j.1834-7819.2011.01294
  3. Kaisarly, D.; El Gezawi, M. Polymerization shrinkage assessment of dental resin composites: A literature review. Odontology 2016, 104, 257–270
  4. Yamauchi, K.; Tsujimoto, A.; Jurado, C. A.; Shimatani, Y.; Nagura, Y.; Takamiza-wa, T.; Barkmeier, W. W.; Latta, M. A.; Miyazaki, M. Etch-and-rinse vs self-etch mode for dentin bonding effectiveness of universal adhesives. J. Oral. Sci. 2019, 61, 549–553. https://doi.org/10.2334/josnusd.18-0433
  5. Jang JH, Lee MG, Woo SU, Lee CO, Yi JK, Kim DS (2016) Comparative study of the dentin bond strength of a new universal adhesive. Dent Mater J 35, 606-612
  6. Tsujimoto A, Barkmeier WW, Takamizawa T, Watanabe H, Johnson WW, Latta MA et al. Simulated localized wear of resin luting cement for universal adhesive systems with different curing mode. J Oral Sci 2018, 60, 29-36
  7. Tsujimoto A, Barkmeier WW, Takamizawa T, Wilwerding T, Latta MA, Miyazaki M (2017) Interfacial characteristics and bond durability on universal adhesives to various substrates. Oper Dent 42, e59-e70.
  8. Nagura Y, Tsujimoto A, Fisher NG, Baruth AG, Barkmeier WW, Takamizawa T et al. (2019) The effect of reduced application time of universal adhesives on enamel bond fatigue durability and surface morphology. Oper Dent 44, 42-53.
  9. Tsujimoto A, Shimatani Y, Nojiri K, Barkmieier WW, Markham MD, Takamizawa T et al. (2019) Influence of surface wetness on bonding effectiveness of universal adhesives in etch-and-rinse mode. Eur J Oral Sci 127, 162-169.
  10. Sano, H.; Takatsu, T.; Ciucchi, B.; Horner, J.A.; Matthews, W.G.; Pashley, D.H. Nanoleakage: Leakage within the hybrid layer. Oper. Dent. 1995, 20, 18–25.
  11. Tsujimoto A, Barkmeier WW, Hosoya Y, Nojiri K, Nagura Y, Takamizawa T et al. (2017) Comparison of bond fatigue durability to enamel of universal adhesives and two-step selfetch adhesives in self-etch mode. Am J Dent 30, 279-284.
  12. Suda S, Tsujimoto A, Barkmeier WW, Nojiri K, Nagura Y, Takamizawa T et al. (2018) Comparison of enamel bond fatigue durability between universal adhesives and two-step self-etch adhesives: effect of phosphoric acid pre-etching. Dent Mater J 37, 244-255
  13. Irmak Ö, Yaman BC, Orhan EO, Ozer F, Blatz MB (2018) Effect of rubbing force magnitude on bond strength of universal adhesives applied in self-etch mode. Dent Mater J 37,139-145.
  14. Tian, F., Zhou, L., Zhang, Z., Niu, L., Zhang, L., Chen, C., Zhou, J., Yang, H., Wang, X., Fu, B., Huang, C., Pashley, D. H., & Tay, F. R. (2016). Paucity of Nanolayering in Resin-Dentin Interfaces of MDP-based Adhesives. Journal of dental research, 95(4), 380–387. https://doi.org/10.1177/0022034515623741
  15. Inoue N, Tsujimoto A, Takimoto M, Ootsuka E, Endo H, Takamizawa T et al. (2010) Surface free-energy measurements as indicators of the bonding characteristics of single-step self-etching adhesives. Eur J Oral Sci 118, 525-530
  16. De Munck, J.; Van Landuyt, K.; Peumans, M.; Poitevin, A.; Lambrechts, P.; Braem, M.; Van Meerbeek, B. A critical review of the durability of adhesion to tooth tissue: Methods and results. J. Dent. Res. 2005, 84, 118–132.
  17. Pashley, D.H.; Tay, F.R.; Breschi, L.; Tjäderhane, L.; Carvalho, R.M.; Carrilho, M.; Tezvergil-Mutluay, A. State of the art etch-and-rinse adhesives. Dent. Mater. 2011, 27, 1–16.
  18. Sun, J. H.; Chen, F.; Kanefuji, K.; Chowdhury, A. F. M. A.; Carvalho, R. M.; Sano, H. Application of a New Microtensile Bond Strength Testing Technique for the Evaluation of Enamel Bonding. Chin. J. Dent. 2021, 24, 159–166. https://doi.org/10.3290/j.cjdr.b1965031
  19. Sebold, M.; Lins, R. B. E.; Sahadi, B. O.; Santi, M. R.; Martins, L. R. M.; Giannini, M. Microtensile Bond Strength, Bonding Interface Morphology, Adhesive Resin Infiltration, and Marginal Adaptation of Bulk-fill Composites Placed Using Different Adhesives. J. Adhes. Dent. 2021, 23, 409–420. https://doi.org/10.3290/j.jad.b2000221
  20. Cuevas-Suárez, C.E.; da Rosa, W.L.O.; Lund, R.G.; da Silva, A.F.; Piva, E. Bonding performance of universal adhesives: An updated systematic review and meta-analysis. J. Adhes. Dent. 2019, 21, 7–26.
  21. Ahmed, M.H.;Yoshihara, K.;Mercelis, B.;Van Landuyt, K.; Peumans M.; Van Meerbeek, B. Quick bonding using a universal adhesive. Clin. Oral. Investig. 2020, 24, 2837–2851.
  22. Rosa, W. L.; Piva, E.; Silva, A. F. (2015). Bond strength of universal adhesives: A systematic review and meta-analysis. J. Dent. 2015, 43, 765–776. https://doi.org/10.1016/j.jdent.2015.04.003
  23. Saikaew, P.; Matsumoto, M.; Chowdhury, A.F.; Carvalho, R.M.; Sano, H. Does shortened application time affect long-term bond strength of universal adhesives to dentin? Oper. Dent. 2018, 43, 549–558.
  24. Perdigão, J. Current perspectives on dental adhesion: (1) Dentin adhesion - not there yet. Jpn. Dent. Sci. Rev. 2020, 56, 190–207.
  25. G2-BOND Universal The new standard of 2-bottle Universal Bonding. Available online: https://europe.gc.dental/en-LB/products/g2bonduniversal (accessed on 10. 9. 2022.)
  26. Van Meerbeek, B.; Yoshihara, K.; Van Landuyt, K.; Yoshida, Y.; Peumans, M. From Buonocore’s pioneering acid-etch technique to self-adhering restoratives. A status perspective of rapidly advancing dental adhesive technology. J. Adhes. Dent. 2020, 22, 7–34.
  27. Blumer,L.; Schmidli,F.; Weiger,R.; Fischer, J. A systematic approach to standardize artificial aging of resin composite cements. Dent. Mater. 2015, 31, 855–863.
  28. ISO 29022:2013. Dentistry—Adhesion—Notched-Edge Shear Bond Strength Test; European Standard: Geneva, Switzerland, 2013.
  29. Ismail, A.M.; Bourauel, C.; ElBanna, A.; Salah Eldin, T. Micro versus Macro Shear Bond Strength Testing of Dentin-Composite Interface Using Chisel and Wireloop Loading Techniques. Dent. J. 2021, 9, 140. https://doi.org/10.3390/dj9120140
  30. Burke, F.J.; Hussain, A.; Nolan, L.; Fleming, G.J. Methods used in dentine bonding tests: An analysis of 102 investigations on bond strength. Eur. J. Prosthodont. Restor. Dent. 2008, 16, 158–165.
  31. O’Donnell, J.N.; Schumacher, G.E.; Antonucci, J.M.; Skrtic, D. Adhesion of amorphous calcium phosphate composites bonded to dentin: A study in failure modality. J. Biomed. Mater. Res. Part B Appl. Biomater. 2009, 90, 238–249.
  32. Perdigão, J. New developments in dental adhesion. Dent. Clin. North. Am. 2007, 51, 333–357.
  33. Braga, R.R.; Meira, J.B.; Boaro, L.C.; Xavier, T.A. Adhesion to tooth structure: A critical review of “macro” test methods. Dent. Mater. 2010, 26, 38–49.
  34. Pecora, N.; Yaman, P.; Dennison, J.; Herrero, A. Comparison of shear bond strength relative to two testing devices. J. Prosthet. Dent. 2002, 88, 511–515.
  35. Yoshihara, K.; Nagaoka, N.; Nakamura, A.; Hara, T.; Yoshida, Y.; Van Meerbeek, B. Nano-Layering Adds Strength to the Adhesive Interface. J. Dent. Res. 2020, 100, 515–521.
  36. Van Landuyt, K.L.; Snauwaert, J.; Peumans, M.; De Munck, J.; Lambrechts, P.; Van Meerbeek, B. The role of HEMA in one-step self-etch adhesives. Dent. Mater. 2008, 24, 1412–1419.
  37. [Ahmed, M.H.; Yoshihara, K.; Yao, C.; Okazaki, Y.; Van Landuyt, K.; Peumans, M.; Van Meerbeek, B. Multiparameter evaluation of acrylamide HEMA alternative monomers in 2-step adhesives. Dent. Mater. 2020, 37, 30–47.
  38. https://europe.gc.dental/hr-HR/products/g2bonduniversal, (accessed on 29. 1. 2023.)
  39. Van Meerbeek, B.; Peumans, M.; Poitevin, A.; Mine, A.; Van Ende, A.; Neves, A.; De Munck, J. Relationship Between Bond-Strength Tests and Clinical Outcomes. Dent. Mater. 2010, 26, 100–121.
  40. Tichy, A.; Hosaka, K.; Yang, Y.; Motoyama, Y.; Sumi, Y.; Nakajima, M.; Tagami, J. Can a New HEMA-free Two-step Self-etch Adhesive Improve Dentin Bonding Durability and Marginal Adaptation? J. Adhes. Dent. 2021, 23, 505–512.
  41. Tsujimoto, A.; Fischer, N. G.; Barkmeier, W. W.; Latta, M. A. (2022). Bond Durability of Two-Step HEMA-Free Universal Adhesive.  J. Funct. Biomater. 2022, 13, 134. https://doi.org/10.3390/jfb13030134
  42. Heintze, S.; Rousson, V.; Mahn, E. Bond strength tests of dental adhesive systems and their correlation with clinical results—A meta-analysis. Dent. Mater. 2015, 31, 423–434.
  43. Burrow, M.F.; Thomas, D.; Swain, M.V.; Tyas, M.J. Analysis of tensile bond strength using Weibull statistics. Biomaterials 2004, 25, 5031–5035.
  44. Lu, C.S.; Danzer, R.; Fischer, F.D. Fracture statistics of brittle materials: Weibull or normal distribution. Phys. Rev. E 2002, 65, 1–4.
  45. Bradna, P.; Vrbova, R.; Dudek, M.; Roubickova, A.; Housova, D. Comparison of bonding performance of self-etching and etch-and-rinse adhesives on human dentin using reliability analysis. J. Adhes. Dent. 2008, 10, 423–429.
  46. Luhrs, A.K.; Guhr, S.; Schilke, R.; Borchers, L.; Geurtsen, W.; Gunay, H. Shear bond strength of self-etch adhesives to enamel with additional phosphoric acid etching. Oper. Dent. 2008, 33, 155–162.
  47. Placido, E.; Meira, J.B.; Lima, R.G.; Muench, A.; de Souza, R.M.; Ballester, R.Y. Shear versus micro-shear bond strength test: A finite element stress analysis. Dent. Mater. 2007, 23, 1086–1092.
  48. Roos, M.; Schatz, C.; Stawarczyk, B. Two independent prospectively planned blinded Weibull statistical analyses of flexural strength data of zirconia materials. Materials 2016, 9, 512.
  49. Blumer,L.; Schmidli,F.; Weiger,R.; Fischer, J. A systematic approach to standardize artificial aging of resin composite cements. Dent. Mater. 2015, 31, 855–863.
  50. Hanabusa, M.; Mine, A.; Kuboki, T.; Momoi, Y.; Van Ende, A.; Van Meerbeek, B.; De Munck, J. Bonding effectiveness of a new “multi-mode” adhesive to enamel and dentine. J. Dent. 2012, 40, 475-484
  51. Leite, M.L.; Costa, C.A.; Duarte, R.M.; Andrade, A.K.; Soares, D.G. Bond strength and cytotoxicity of a universal adhesive according to the hybridization strategies to dentin. Braz. Dent. J. 2018, 29, 68-75.
  52. Tsujimoto A, Barkmeier WW, Hosoya Y, Nojiri K, Nagura Y, Takamizawa T et al. (2017) Comparison of bond fatigue durability to enamel of universal adhesives and two-step selfetch adhesives in self-etch mode. Am J Dent 30, 279-284.
  53. Suda S, Tsujimoto A, Barkmeier WW, Nojiri K, Nagura Y, Takamizawa T et al. (2018) Comparison of enamel bond fatigue durability between universal adhesives and two-step self-etch adhesives: effect of phosphoric acid pre-etching. Dent Mater J 37, 244-255

Comment 5:   In general, the article is interesting and useful, perhaps the authors lacked microstructural studies. Authors should consider supplementing their SEM study with an analysis or other description of the structure of the resulting materials. After correcting all the comments, the article should be sent for re-review for its further evaluation.

Response: Thank you for your comment. We appreciate your suggestion about adding microstructural studies to the article. Unfortunately, we were unable to include SEM analysis in this study. However, we did perform an examination of the fragments and dentin surface using a 3.6x optical loupe (Carl Zeiss Meditec AG, Oberkochen, Germany) and Dinolite microscope to determine the type of fracture, i.e., the cause of failure. The information has been added to both the "Materials and Methods" section and the "Results" section, with the appropriate changes highlighted in green. Additionally, new figures have been included in the manuscript.

“The fractured fragments were examined with a 3.6x optical loupe (Carl Zeiss Meditec AG, Oberkochen, Germany) and DinoLite microscope (DinoLite, Almere, The Netherlands) to determinethe type of fracture, i.e., the cause of failure. If the fracture line is between the tooth and the composite cylinder, the fracture mode is classified as adhesive. The fracture mode is classified as mixed if the fracture line runs partially along the adhesive interface and penetrates one of the substrates, so we distinguish between the mixed fracture mode in the dentin or in the composite (depending on which substrate it covers). If more than 75% of the adhesive area involves either dentin or composite, the fracture mode is classified as cohesive. Surface morphology was examined by three examiners (E.K., S.B., and S.J.K) using different magnifications up to 200 magnification and photomicrographs of representative areas were taken.

3.1. Type of fractures

3.1. Type of fractures

No significant differences were found between fracture types. All fractures, i.e. fracture lines, were located between the tooth and the composite cylinder, so the fracture mode was classified as adhesive (Figure...). None of the sepeciments were classified as mixed or cohesive.”

Once again, we thank you for your invaluable input.

Best regards,

Authors

Round 2

Reviewer 4 Report

Thank you for the revisions.

Reviewer 5 Report

All my comments were taken into account and appropriate corrections were done. The article looks much better.

I recommend the article for publishing.